

**Counter-rotating eddy pair in the Luzon Strait**
Sun Ruili[1,2*], Li Peiliang[1,3], Gu Yanzhen[3], Zhai Fangguo[4], Yan Yunwei[2], Li Bo[5]
[1]Hainan Institute of Zhejiang University, Sanya, Hainan, China
[2]State Key Laboratory of Satellite Ocean Environment Dynamics, Second Institute of
Oceanography, Ministry of Natural Resources, Hangzhou, China
[3]Ocean college, Zhejiang University, Zhoushan, China
[4]College of Oceanic and Atmospheric Sciences, Ocean University of China, Qingdao, China
[5]State Key Laboratory of Tropical Oceanography, South China Sea Institute of Oceanology,
Chinese Academy of Sciences, Guangzhou, China
Corresponding author:Sun Ruili, sunruili2007@126.com
**Abstract:**
Based on satellite remote sensing observation data and Hybrid Coordinate Ocean Model
(HYCOM) reanalysis data, we studied the counter-rotating eddy pair in the Luzon Strait (LS).
Statistical analysis revealed that when an anticyclonic mesoscale eddy (AE) (cyclonic mesoscale
eddy (CE)) in the Northwest Pacific (NWP) gradually approached the east side of the LS, a CE (an
AE) gradually formed on the west side of the LS, and it was defined as the AE (CE) mode of the
counter-rotating eddy pair in the LS. The counter-rotating eddy pair exhibited obvious seasonal
variation: the AE mode mainly occurred in the summer half of the year, while the CE mode
mainly occurred in the winter half of the year. The mean durations of the AE mode and CE mode
were both about 70 days. Based on energy analysis and the vorticity budget equation, the dynamic
mechanism of the counter-rotating eddy pair occurrence was determined to be as follows: the AE
(CE) on the east side of the LS causes a positive (negative) vorticity anomaly through horizontal

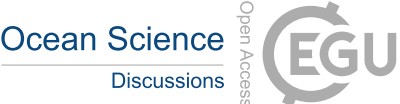

velocity shear on the west side of the LS and the positive (negative) vorticity anomaly is
transported westward by the zonal advection of the vorticity, finally leading to the formation of the
CE (AE) on the west side of the LS.
**Keywords:** counter-rotating eddy pair; Luzon Strait; vorticity budget equation; barotropic
instability;
**1 Introduction**
The Luzon Strait (LS), located between the Taiwan Island and the Luzon Island, is an important
gap for material and energy exchange between the South China Sea (SCS) and the Northwest
Pacific (NWP). The topography around the LS is very complicated. The LS comprises three straits
from north to south: the Bashi Strait, the Balintang Strait and the Babuyan Strait. The Batanes
Islands and Babuyan Islands are located in these straights (Figure 1). These complex topographic
features can significantly affect the dynamic ocean process around the LS and play an important
role in the material and energy exchange between the SCS and the NWP (Lu and Liu, 2013; Sun et
al., 2016a). Sun et al (2016a) pointed out that the Kuroshio bifurcated into west and east branches
when it encounters the Batanes Islands in the LS. The bifurcation of the Kuroshio can significantly
alter the transport of the Kuroshio's main axis, and therefore, it has a potential impact on the
intrusion of the Kuroshio into the SCS. The bifurcation of the Kuroshio is also affected by
mesoscale eddies (Sun et al., 2016a).
Mesoscale eddies widely exist in the vicinity of the LS, and many of them come from the
NWP. These mesoscale eddies from the NWP can carry an enormous amount of kinetic energy and
can alter the local circulation, including the Kuroshio. Some of them cross the LS into the SCS,
thus contributing to the material and energy exchange between the SCS and the NWP. However, in



addition to their method of entering the SCS, it is important to determine if the mesoscale eddies
from the NWP affect the material and energy exchange between the SCS and WNP in other ways?
For example, mesoscale eddies from the NWP do not have to enter into the SCS, and they can
affect the SCS circulation through eddy-eddy interactions.

Numerous studies have been conducted on the eddy-eddy interactions in the vicinity of the

LS. Jing and Li (2003) used satellite remote sensing observation data to discover a cyclonic
mesoscale cold eddy around the Lanyu Island to the northeast of the LS, and they speculated that
the overshooting of the meandering Kuroshio when it leaves the SCS and the effects of the
conservation of the potential vorticity may be the formation mechanism of the Lanyu cold eddy.
Using a neural network and satellite remote sensing observation data, Yin et al. (2014) statistically
demonstrated that a CE (AE) from the NWP can decrease (increase) the velocity of the Kuroshio,
thus causing the Kuroshio to intrude into the East China Sea (ECS) in a stronger (weaker) cyclonic
manner. This showed that mesoscale eddies from the NWP can alter the ECS circulation without
entering the ECS. Sun et al (2016b) used updated satellite remote sensing observation data and
composition analysis to determine that the combined action of the Kuroshio loop and an AE from
the NWP led to the formation of the Lanyu cold eddy to the northeast of the LS. However, none of
these studies determined the dynamic mechanism of eddy-eddy interactions. Based on satellite
observation data, in situ observation data and numerical modelling data, Zhang et al. (2007)
analyzed the energy budget of the Kuroshio invading the SCS, and they determined that the
northern branch of the anticyclonic circulation caused by the Kuroshio loop has a large horizontal
shear stress and thus leads to the formation of a CE southwest of the Taiwan Island through the
barotropic instability, which proposed a dynamic mechanism for eddy-eddy interactions around



the LS.

Although some research related to eddy-eddy interactions in the vicinity of the LS has been

conducted, and it has been discovered that mesoscale eddies are widely distributed on the west and
east sides of the LS (Figure 2), it is not clear whether the mesoscale eddies on the west and east
sides of the LS can interact and exchange energy between the SCS and the NWP. In order to
explore this issue, we compared the sea surface height anomaly (SSHA) distributions when a CE
occurred and when an AE occurred on the east side of the LS (Figure 3). The specific process will
be described in detail in Section 3.1. Figure 3 shows that when an AE (a CE) occurred on the east
side of the LS, a CE (an AE) formed on the west side of the LS, which was observed in the in situ
observation data (Huang et al., 2019). This is referred to as the counter-rotating eddy pair
phenomenon in this paper. This counter-rotating eddy pair process inevitably led to energy
exchange between the SCS and the NWP. To the best of our knowledge, it is not only a new
phenomenon proposed for the first time, but it is also a new mechanism proposed for the first time,
i.e., that energy is exchanged between the SCS and the NWP. We analyzed the statistical
characteristics and dynamic mechanism of this phenomenon. The rest of this paper is organized as
follows. Section 2 briefly introduces the data and methods. Section 3 presents the research results.
Section 4 presents the discussion and conclusion.



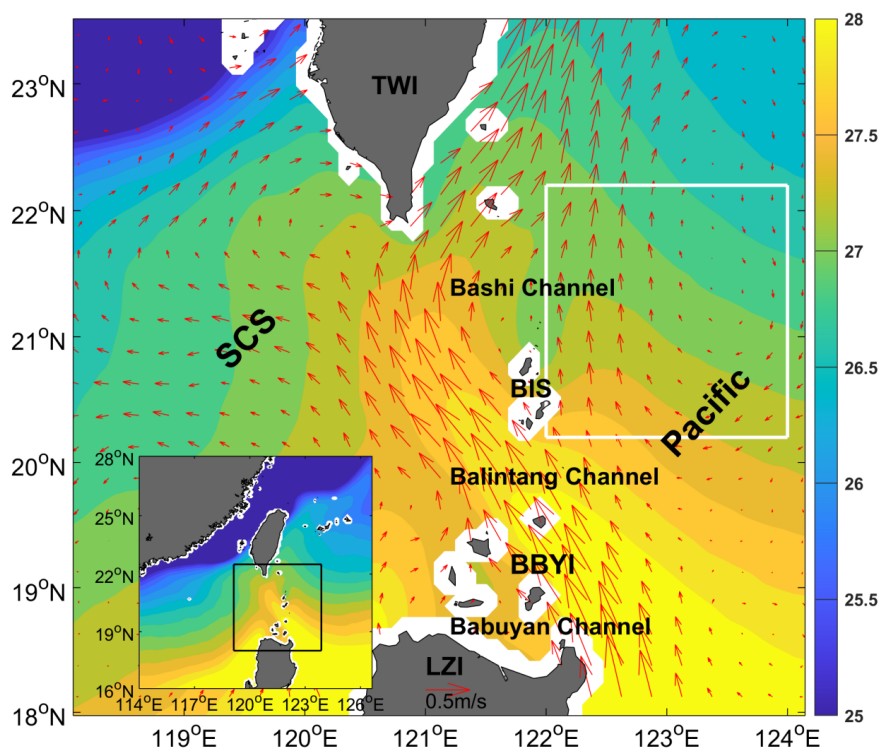

Figure 1. Spatial distribution of the RSS SST (℃; shading) and CMEMS geostrophic current (m/s;

vectors) from 2003 to 2020. SCS: South China Sea; BIS: Batanes Islands; TWI: Taiwan Island;

LZI: Luzon Island; BBYI: Babuyan Islands. The white box borders 20.2-22.2°N, 122-124°E. The

extent of the main map is shown as a black box in the inset.



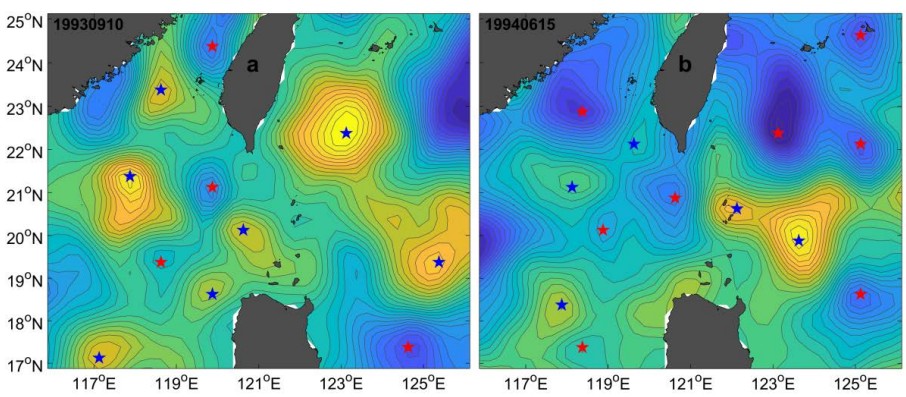

Figure 2. Spatial distribution of the CMEMS SSHA and locations of centers of the eddies on (a)

September 10, 1993 and (b) June 15, 1994. The colors and contours represent the SSHA. The blue

star and red star denote the locations of the AE and CE, respectively.

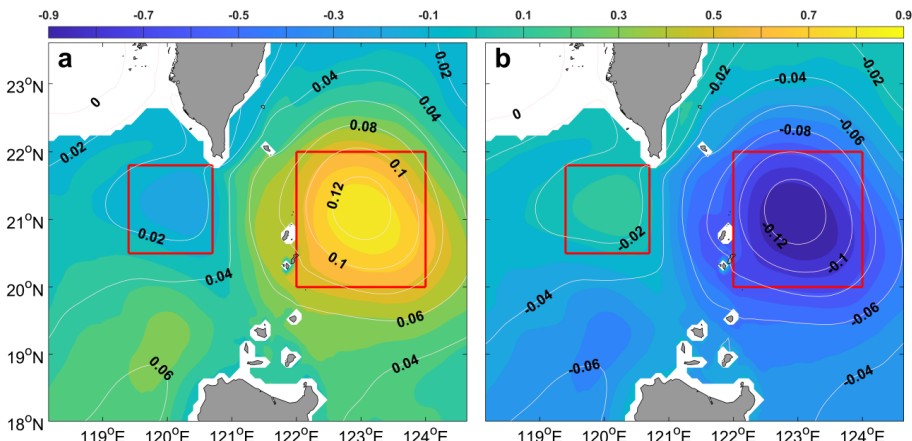

Figure 3. Spatial distribution of the counter-rotating eddy pair in the LS. (a) Spatial pattern of the

AE mode; and (b) Spatial pattern of the CE mode. The contours represent the SSHA (unit: m). The

colors represent the sea temperature anomaly (unit:℃) at a depth of 300 m. The interval of the

SSHA is 0.03 m. The red boxes on the west side and east side of the LS border 20.5-21.8°N,

119.4-120.7°E and 20-22°N, 122-124°E, respectively. This figure is similar to Figure 3 in Sun et

al (2018), and is based on HYCOM data.



**2 Data and methods**

2.1 Data

Satellite remote sensing SSHA, geostrophic current and geostrophic current anomaly data are

provided by the Copernicus Marine Environment Monitoring Service (CMEMS) (the download

website:     https://resources.marine.copernicus.Eu/?option=com_csw&view=details&product_id=

SEALEVEL_GLO_PHY_L4_REP_OBSERVATIONS_008_047). The data set is generated by

the processing system including data from all altimeter missions: Sentinel-3A/B, Jason-3, HY-2A,

Cryosat-2, OSTM/Jason-2, Jason-1, Topex/Poseidon, Envisat, GFO, ERS-1/2. The data set

provides a global coverage data from 1 January 1993 to present, with a spatial resolution of

$0.25°×0.25°$ and temporal sampling frequency of 1 day. It provides one near-real time component

and one delayed time component. The delayed time component has been inter-calibrated and

provided a homogeneous and highly accurate long time series of all altimeter data (Pujol and

Francoise, 2019), and is chosen in this paper.

Hybrid Coordinate Ocean Model (HYCOM) organization provided the HYCOM reanalysis

data (the download website: https://www.hycom.org/dataserver/gofs-3pt0/reanalysis). The data set

is based on ocean prediction system output, and the product with the longest time span from 2

October 1992 to 31 December 2012, is chosen among HYCOM data-assimilation product

provided by the HYCOM organization. The data set is based on ocean prediction system output

with a spatial resolution of $0.08°×0.08°$ and 40 standard z-levels between 80.48°S and 80.48°N. It

provides temperature, salinity, sea surface height, zonal flow and meridional flow.

The data set of wind was provided by the National Climate Data Center (NCDC) (https://

www.ncdc.noaa.gov/data-access/marineocean-data/blended-global/blended-sea-winds). The data





set merges multiple satellite observation, in-situ instrument and related individual products. It
provides 6-hours, daily, monthly and climate data with a spatial resolution of 0.25°×0.25°. The
data set contains globally gridded ocean surface vector winds and wind stresses (Zhang et al.,

2006).

Sea surface temperature (SST) data comes from Remote Sensing System (RSS; The download
websites: http://www.remss.com/measurements/sea-surface-temperature/). The data set merges
near-coastal capability and high spatial resolution of the infrared SST data with through-cloud
capabilities of the microwave SST data, and has applied atmospheric corrections. It provides daily
data with a spatial resolution of 9km×9km from July 1, 2002 to present.
2.2 Methods
2.2.1 Eddy energetic and hydrodynamic instability formula
The formation mechanisms of mesoscale eddies in the ocean are commonly attributed to
baroclinic and barotropic instabilities (Pedlosky, 1987; Zhang et al., 2015; Zhang et al., 2017). The
barotropic conversion (BT) and the baroclinic conversion (BC) are manifestations of the
baroclinic and barotropic instabilities, respectively, and they are the major eddy energy sources
around the LS (Yang et al., 2013; Zhang et al., 2013, 2015, 2017). In addition, the wind stress
work (WW) can also contribute to the formation of eddies (Ivchenko, 1997; Sun et al., 2015). The
BT, BC, and WW can be expressed as follows (Ivchenko, 1997; Oey, 2008):
$$BT = -\int \left( \overline{u'^2} \frac{\partial \overline{u}}{\partial x} + \overline{v'^2} \frac{\partial \overline{v}}{\partial y} + \overline{u'v'} \frac{\partial \overline{u}}{\partial y} + \overline{u'v'} \frac{\partial \overline{v}}{\partial x} \right) dz, \qquad (1)$$

$$BC = -\int \frac{g^2}{\rho_0^2 N^2} \left( \overline{u'\rho'} \frac{\partial \overline{\rho}}{\partial x} + \overline{v'\rho'} \frac{\partial \overline{\rho}}{\partial y} \right) dz, \qquad (2)$$

$$WW = \frac{1}{\rho} \left( \overline{u'\tau'_x} + \overline{v'\tau'_y} \right), \qquad (3)$$

Where $t$ is the time; $u, v,$ and $w$ are the zonal velocity, meridional velocity and vertical





velocity, respectively, and their positive directions are east, north and up, respectively. $g$ is the
acceleration due to gravity; $N$ is the buoyancy frequency; $\rho$ is the density of sea water; $\rho_0 =$
$1030 kg \cdot m^{-3}$ is the mean sea water density; $p$ is the sea pressure; and $\tau_x$ and $\tau_y$ are the zonal
and meridional components of the wind stress, respectively. $x, y,$ and $z$ are the conventional
east-west, north-south and up-down Cartesian coordinates, respectively. The depth integrals for
BT and BC are from 400m to the sea surface. The overbar denotes time averaged over an n-day
period, the primes denote deviations from the average, and the other symbols and notations are
standard. The n-day period was chosen to be 70 days according to the Figures 7 and 9, which
show that the period of the counter-rotating eddy pair phenomenon is close to 70 days. BT and BC
were calculated from the HYCOM data. CMEMS surface current velocity data and NCDC wind
data were used to calculate the WW.

2.2.2 Vorticity budget equation

To examine the influence of the vorticity change, we applied the vorticity budget equation:

$$\frac{\partial \zeta}{\partial t} = -u\frac{\partial \zeta}{\partial x} - v\frac{\partial \zeta}{\partial y} - (\zeta + f)\nabla \bullet \vec{u} - v\frac{\partial f}{\partial y} + \frac{1}{\rho^2}\left(\frac{\partial \rho}{\partial x}\frac{\partial P}{\partial y} - \frac{\partial \rho}{\partial y}\frac{\partial P}{\partial x}\right) - \upsilon\frac{\partial^2 \zeta}{\partial^2 z}, \qquad (4)$$

Where $\zeta = \frac{\partial v}{\partial x} - \frac{\partial u}{\partial y}$ is the relative vorticity; $t$ is the time; $u$ and $v$ is the zonal velocity,

meridional velocity, respectively. $x, y,$ and $z$ are the conventional east-west, north-south and
up-down cartesian coordinates, respectively; $f$ is the Coriolis parameter; $\rho$ is the sea water
density; $P$ is the sea water pressure; and $\upsilon$ is the kinematic viscosity coefficient. The items on the
right side of the equation are the zonal advection term, the meridional advection term, the
stretching term, the beta term, the baroclinic term and the diffusion term in turn.
**3 Results**
3.1 Identification of and temporal variation in the counter-rotating eddy pair in the LS



Based on cluster analysis, which is the same as the clustering method used by Sun et al. (2018),
we determined the SSHA and sea temperature anomaly (STA) based on the days when an AE and
a CE existed on the east side and west side of the LS (shown in the white box in Figure 1),
respectively. Figure 3 shows that when an AE (a CE) occurred on the east side of the LS, a CE (an
AE) formed on the west side of the LS, which is defined as a counter-rotating eddy pair in the LS
in this paper. Figure 3a shows that the SSHA in the red box on the east side of the LS inreased
from the outside to the inside, which means that there was an AE. Due to the geostrophic balance
and mass conservation, the AE causes convergence of the sea water, leading to downwelling in its
center, subsequently leading to an increase in the temperature in the deep ocean. This is verified
by the fact that the STA in the red box on the east side of the LS is gradually increases from
outside to inside and the value is the highest in the center. In addition, the SSHA in the red box on
the west side of the LS decreases from outside to inside and the STA is negative, indicating the
presence of a weak CE. Figure 3b is similar to Figure 3a, but for a CE and an AE on the east and
west sides of the LS, respectively.
In order to better reflect the intensity of this phenomenon, we constructed an index which is
defined as the time series of the SSHA in the red box on the east side of the LS minus that on the
west side of the LS, in order to obtain a time series (Figure 4a). We constructed the SSHA based
on the days when the positive and negative intensity index values were more than one standard
deviation away from the mean. An AE (a CE) on the east side of the LS and a CE (an AE) on the
west side of the LS are shown in Figure 4b (Figure 4c) for the days with positive (negative)
intensity index values, which reflects the phenomenon of a counter-rotating eddy pair in the LS
well. The pattern in Figure 4b (Figure 4c) is defined as the AE (CE) mode of the counter-rotating


eddy pair in this paper.

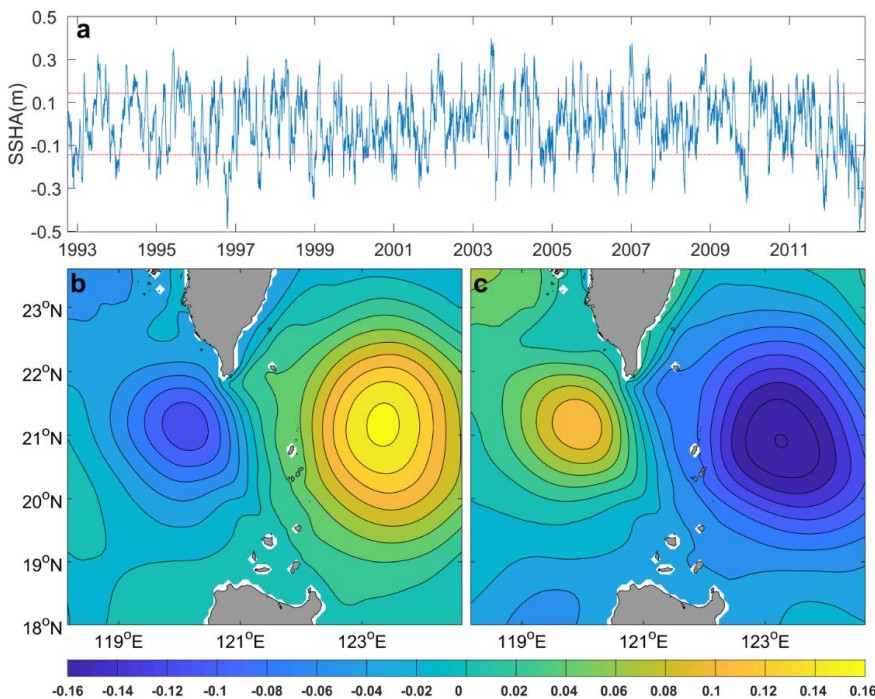

Figure 4. (a) Time series of the intensity index of the counter-rotating eddy pair in the LS. The red
dotted line above (below) represents the sum (difference) of one time the standard deviation and
the average value of the time series. Composition of the SSHA for (b) the positive intensity index
days and (c) the negative intensity index days. The interval of the SSHA is 0.02 m. This figure is
based on HYCOM data.

We counted the temporal distribution of the positive and negative intensity index values.

Figure 5a (Figure 5b) shows that most of the AE (CE) mode of the instances of the
counter-rotating eddy pair occurred in the summer (winter) half of the year. The first two months
with the highest incidences of the AE (CE) mode occur were May and June (December and
January), and their occurrence rates were 17.01% and 15.47% (19.69% and 15.57%), respectively.
We constructed the geostrophic current in May and June (Figure 6a) and in December and January





(Figure 6b). The patterns of the Kuroshio in Figures 6a and 6b exhibit as the "Leap" and "Loop"
patterns of the Kuroshio in the LS, which illustrates that the Leap and Loop patterns of the
Kuroshio contribute to the occurrence of the AE mode and the CE mode of the counter-rotating
eddy pair, respectively. Figure 6c (Figure 6d) shows that the geostrophic current anomaly in the
northern LS is northward (southward). It produces produce positive (negative) vorticity through
horizontal velocity shear on the west side of the LS, and then contributes to the formation of a CE
(an AE) on the west side of the LS. We will discuss the dynamic mechanism of the
counter-rotating eddy pair phenomenon in detail in Section 3.3.

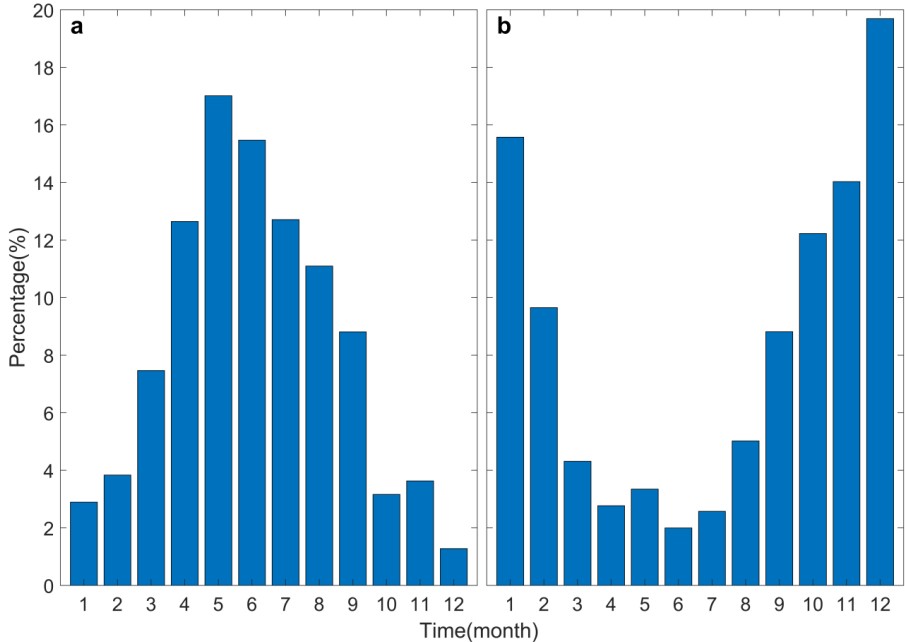


Figure 5. Seasonal distribution of the occurrence rate for (a) the positive intensity index and (b)
the negative intensity index.

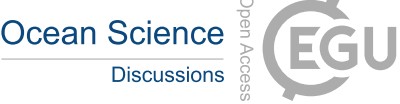


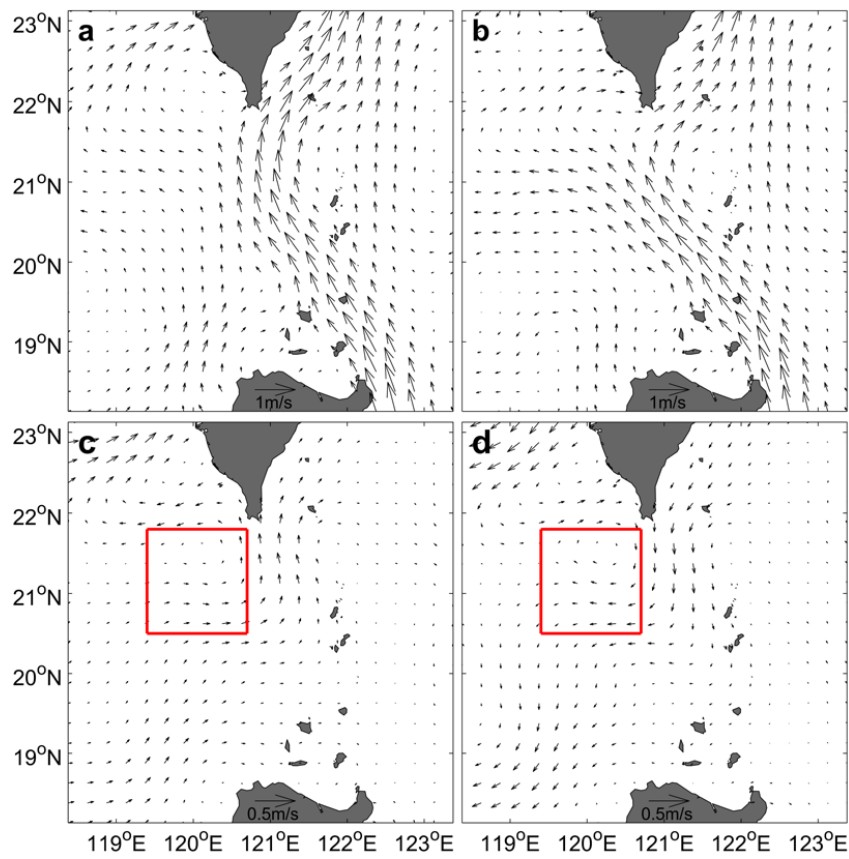

Figure 6. Climatic distribution of geostrophic current in (a) May and June; and (b) December and
January; Climatic distribution of the geostrophic current anomaly in (c) May and June; and (d)
December and January. The red boxes in the (c) and (d) outline 20.5-21.8°N, 119.4-120.7°E,
which represents the position of the mesoscale eddies on the west side of the LS. The figure is
based on CMEMS data.
3.2 Evolution of the counter-rotating eddy pair in the LS
Figure 7 shows the spatial evolution of the AE mode of the counter-rotating eddy pair in the
LS. It shows that at the beginning, for example, at t = -24, there was a weak AE far away from the
east side of LS, but there was no CE on the west of the LS. From t = -20 to t = 0, as the AE in the



NWP approached the northern LS, a CE gradually formed on the west of the LS. At t = 0, the AE
mode reached the pinnacle. Then from t = 4 to t = 36, as the AE in the NWP gradually moved
away from to the northern LS, the CE on the west of the LS gradually weakened until it finally
died out.

The growth and weakening of a mesoscale eddy must be accompanied by a change in its

relative vorticity. Figure 8a shows that as the AE on the east side of the LS approached and then
moved away from the northern LS, its relative vorticity initially decreased first and increased,
while the relative vorticity of the corresponding CE on the west side of the LS initially increased
and then decreased. The maximum negative (positive) value of the time series of the AE (CE) on
the east (west) side of the LS can reached -4.2 s$^{-1}$ (3.6 s$^{-1}$). These time series had a good
correspondence, and their correlation coefficient was -0.97 at the 95% confidence level. Therefore,
the temporal variations in the relative vorticity in Figure 8a verify the evolution of the AE mode of
the counter-rotating eddy pair in the LS.

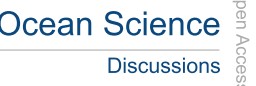


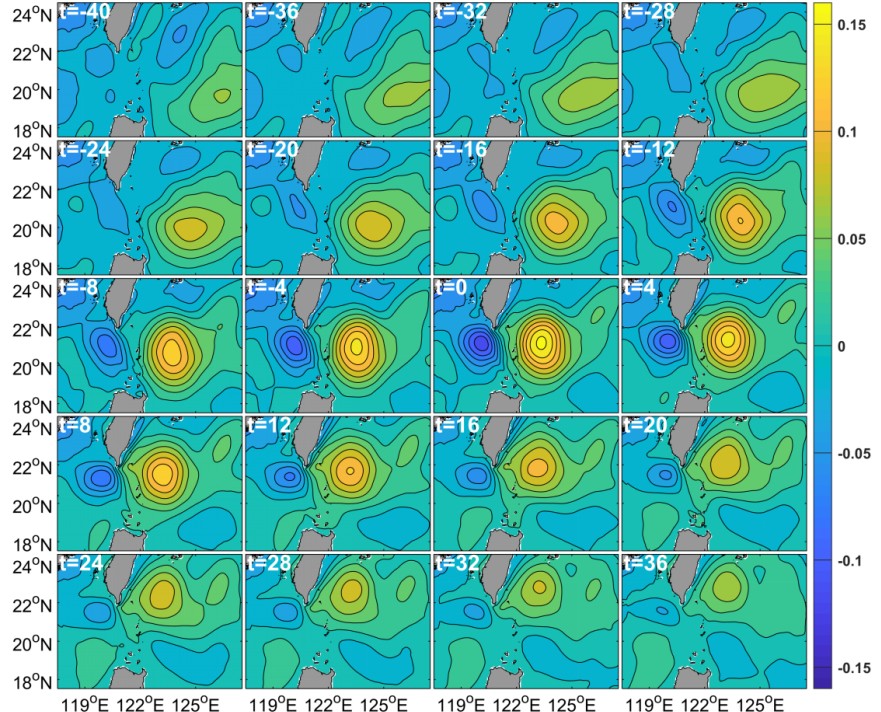


Figure 7. Evolution of the AE mode of the counter-rotating eddy pair in the LS based on HYCOM

data. The contours and shading both represent the SSHA (unit: m). The interval of the SSHA is

0.02 m. The t in the top-left corner of each panel denotes the days before (negative value) or after

(positive value) the AE mode of the counter-rotating eddy pair reached the pinnacle (t=0). t = 0

corresponds to the time of the Figure 4b.



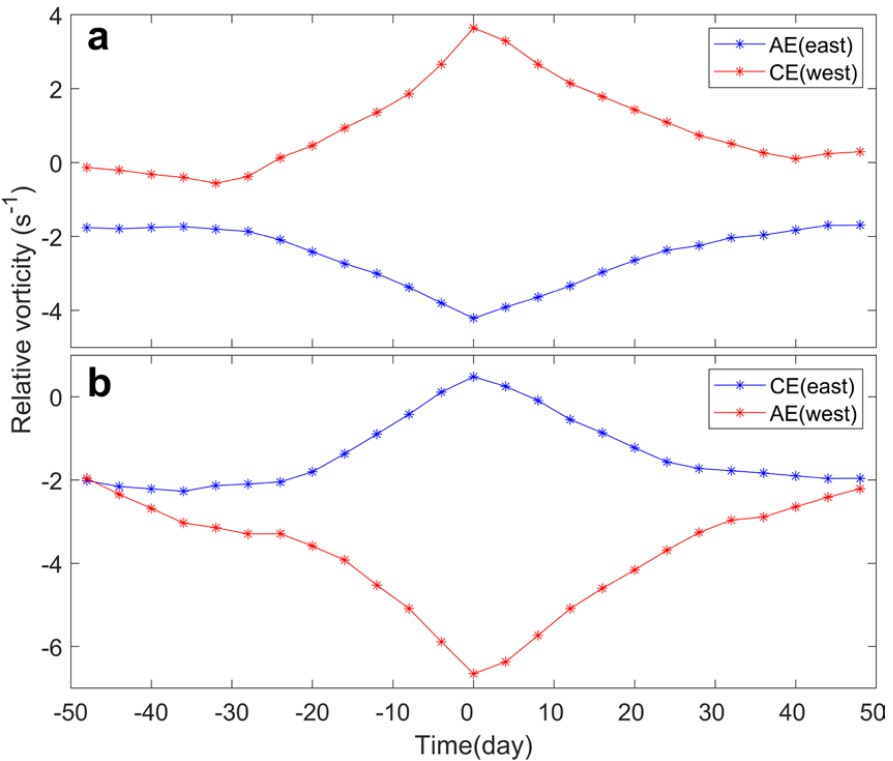

Figure 8. The distribution of the relative vorticity surrounded by the red boxes in Figure 3. (a)

Relative vorticity of the AE mode over time. The blue (red) line represents the time series of the

relative vorticity of the AE (CE) on the east (west) side of the LS, which corresponds to Figure 7;

(b) Relative vorticity of the CE mode over time. The blue (red) line represents the time series of

the relative vorticity of the CE (AE) on the east (west) side of the LS, which corresponds to Figure

9. This figure is based on HYCOM data.

Figure 9 is the same as Figure 7 but for the CE mode of the counter-rotating eddy pair in the

LS. It shows that at the beginning, for example, at t = -32, there was a weak CE far away from the

east side of the LS, but there was no an AE on the west of the LS. From t = -28 to t = 0, as the CE

in NWP approached the northern LS, an AE gradually formed on the west side of the LS. At t = 0,

the CE mode of the evolution of the counter-rotating eddy pair reached the pinnacle. Then from t



= 4 to t = 36, as the CE in the NWP gradually moved away from the northern LS, the AE in the
west side of the LS gradually weakened until it finally died out. Figure 8b is the same as Figure 8a
but for the CE mode. It shows that as the CE on the east side of the LS approached and moved
away from the northern LS, its relative vorticity initially increased and then decreased; while the
relative vorticity of the corresponding AE on the west side of the LS initially decreased and then
increased. The maximum positive (negative) value of the time series of the CE (AE) on the east
(west) side of the LS can reach $0.48\ s^{-1}$ ($-6.7\ s^{-1}$). These time series had a good correspondence and
their correlation coefficient was -0.96 at the 95% confidence level. Therefore, the temporal
variations in the relative vorticity in Figure 8b verify the evolution of the AE mode of the
counter-rotating eddy pair in the LS. The evolution of the AE (CE) mode of the counter-rotating
eddy pair in the LS is also reflected by the satellite observations (Figures 10 and 11).





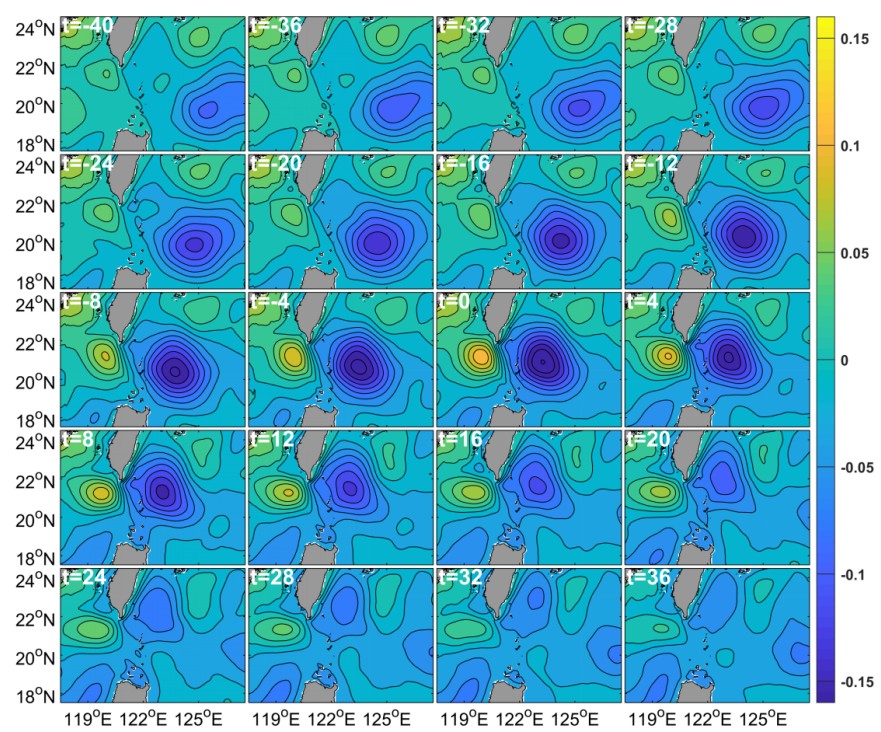


Figure 9 is the same as Figure 7 but for the CE mode of the counter-rotating eddy pair in the LS.




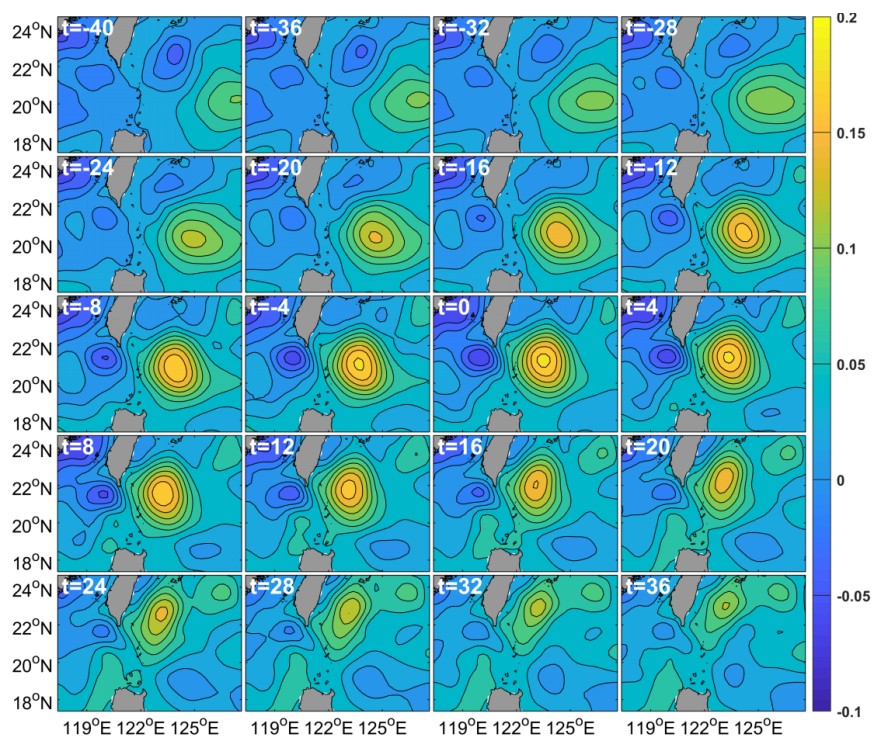


Figure10 is the same as Figure 7, but it is based on the CMEMS data.





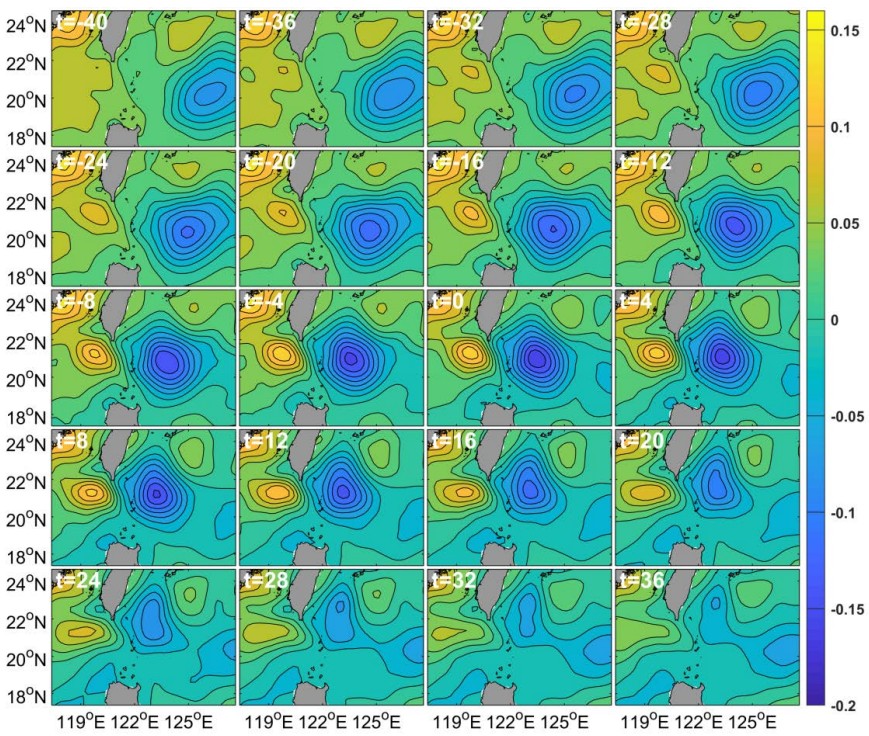


Figure 11 is the same as Figure 9, but it is based on the CMEMS data.

3.3 Formation mechanism of the counter-rotating eddy pair in the LS

Zhang et al. (2017) reported that CEs form mainly due to the barotropic instability caused by

horizontal velocity shear of the Kuroshio Loop current southwest of the Taiwan Island. Huang et
al. (2019) discovered that an AE from the NWP caused a CE to form on the west side of the LS
via horizontal velocity shear. In addition, Figures 4b and 4c show that the dense contour of the
SSHA means that there were strong current anomalies and thus strong horizontal velocity shear at
the junction of the AE and CE. Therefore, we investigated the role of horizontal velocity shear in
the formation of a counter-rotating eddy in the LS.

Because meridional horizontal velocity shear is weak, we only show the zonal velocity shear.

Figure 12 shows that from t = -40 to t = 0, as the AE on the east side of the NWP gradually



approached the northern LS, the absolute value of the zonal horizontal velocity shear ( $\frac{\partial v}{\partial x}$ )
gradually increased, and a CE gradually formed and strengthened on the west side of the LS. From
t = 0 to t = 36, as the AE gradually moved away from the northern LS, the absolute value of the
zonal horizontal velocity shear gradually decreased, and the CE on the west side of the LS
gradually weakened. Figure 13 is the same as Figure 12, but for the CE mode of the
counter-rotating eddy pair in the LS. It shows a similar corresponding evolution process. This
demonstrates that there is a good correspondence between the zonal horizontal velocity shear and
the evolution process of the counter-rotating eddy pair.

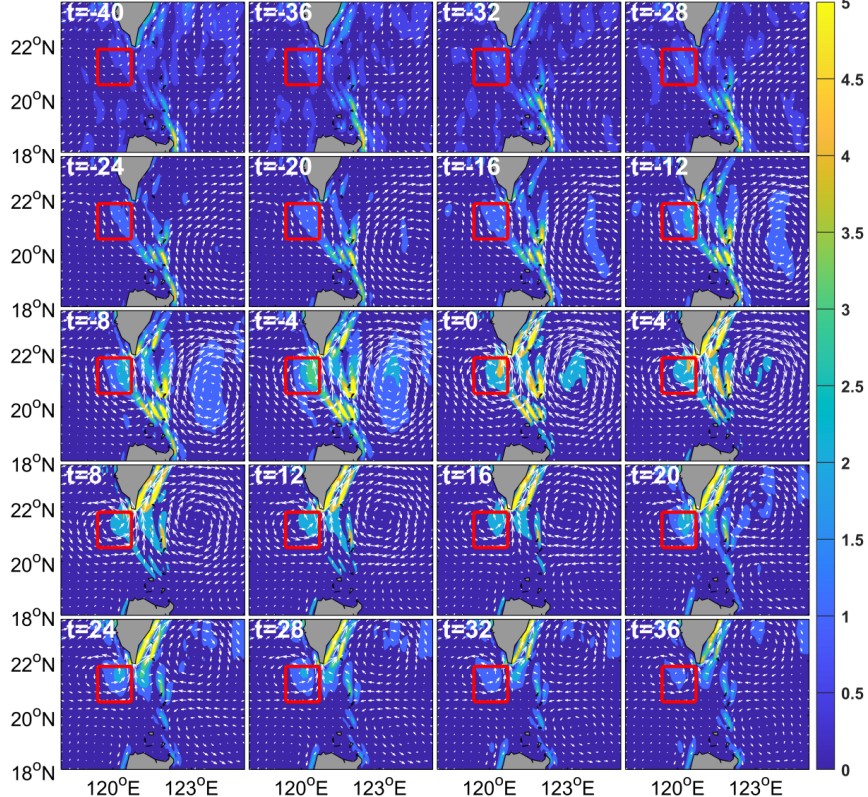


Figure 12 Evolution process of the absolute value of the zonal horizontal velocity shear ( $\frac{\partial v}{\partial x}$ ) for
the AE mode of the counter-rotating eddy pair in the LS based on HYCOM data. The shading



represents the zonal horizontal velocity shear (unit: $10^6$ s$^{-2}$). The vector represents the current
anomaly. The t in the top-left corner of each panel denotes the days before (negative value) or
after (positive value) the AE mode of the counter-rotating eddy pair reached the pinnacle (t = 0).
Time t = 0 corresponds to the time of the Figure 4b. The red boxes on the west side of the LS
cover 20.5-21.8°N, 119.4-120.7°E and represents the location of the CE on the west side of the
LS.

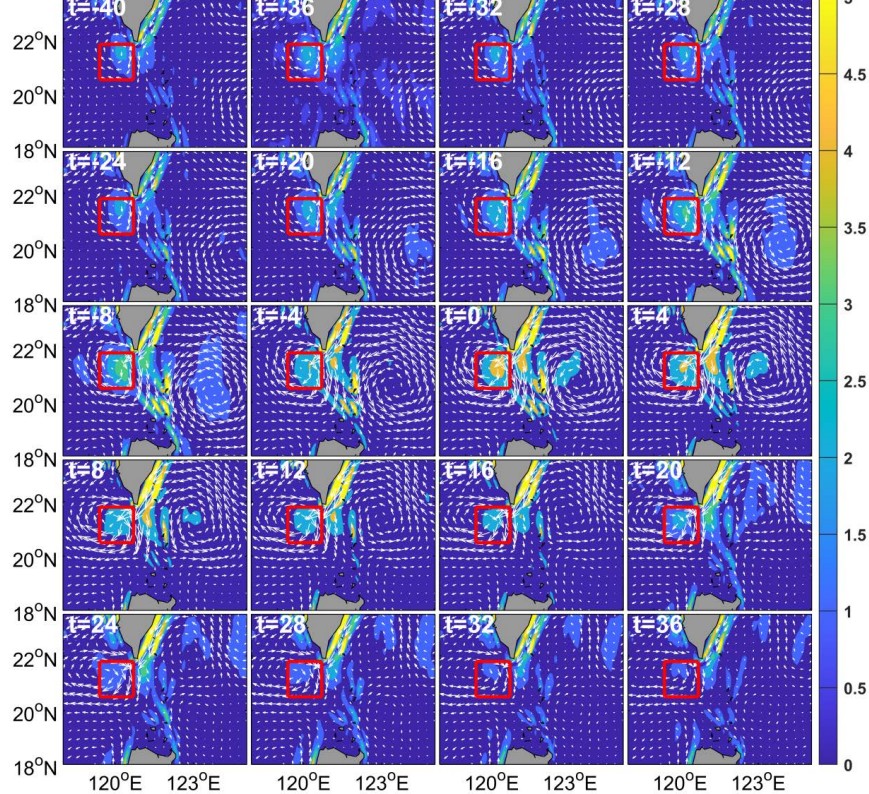

Figure 13 is the same as Figure 12 but for the CE mode of the counter-rotating eddy pair in the
LS.

However, Figure 12 (Figure 13) shows that zonal horizontal velocity shear only occurred on the

right side of the red box, that is, on the right side of the CE (AE). How does the horizontal



velocity shear pass to the entire CE (AE)? To answer this question, we used the vorticity budget
equation. Figures 14 (a1 - f1) are for the AE mode of the counter-rotating eddy pair and show the
contributions of the zonal advection term, meridional advection term, stretching term, beta term,
baroclinic term and diffusion term of the vorticity budget equation, respectively. Compared to the
stretching term, the beta term, baroclinic term and diffusion term, the values of the zonal
advection term and the meridional advection term in the red box is large. However, most of the
values of the meridional advection term in the red box are negative. Only positive vorticity
advection can lead to CE formation, which suggests that the zonal advection term is the main
cause of the CE formation in the red box. To further test this conclusion, Figure 15a shows the
correspondence between the relative vorticity anomaly and the zonal advection of the vorticity in
the red box in Figure 14. It shows that there is a good correspondence and their correlation
coefficient is as high as 0.96 at the 95% confidence level. Therefore, we conclude that the zonal
advection term plays the most important role in the vorticity transport and contributes to the
formation of the CE on the west side of LS.
Figures 14 (a2 - f2) are the same as Figures 14 (a1 - f1), but for the CE mode of the
counter-rotating eddy pair. Figures 14 (a2 - f2) also show that, compared to the stretching term,
beta term, baroclinic term, and diffusion term, the values of the zonal advection term and the
meridional advection term in the red box are large. However, most of the values of the meridional
advection term in the red box are positive. Only negative vorticity advection can lead to AE
formation, which implies that the zonal advection term is the main cause of the AE formation in
the red box. To further test this conclusion, Figure 15b shows the correspondence between the
relative vorticity anomaly and the zonal advection of vorticity. It shows that there is a good





correspondence and their correlation coefficient is as high as 0.84 at the 95% confidence level.
Therefore, we conclude that the zonal advection term plays the most important role in the vorticity
transport and contributes to the formation of the AE on the west side of the LS.

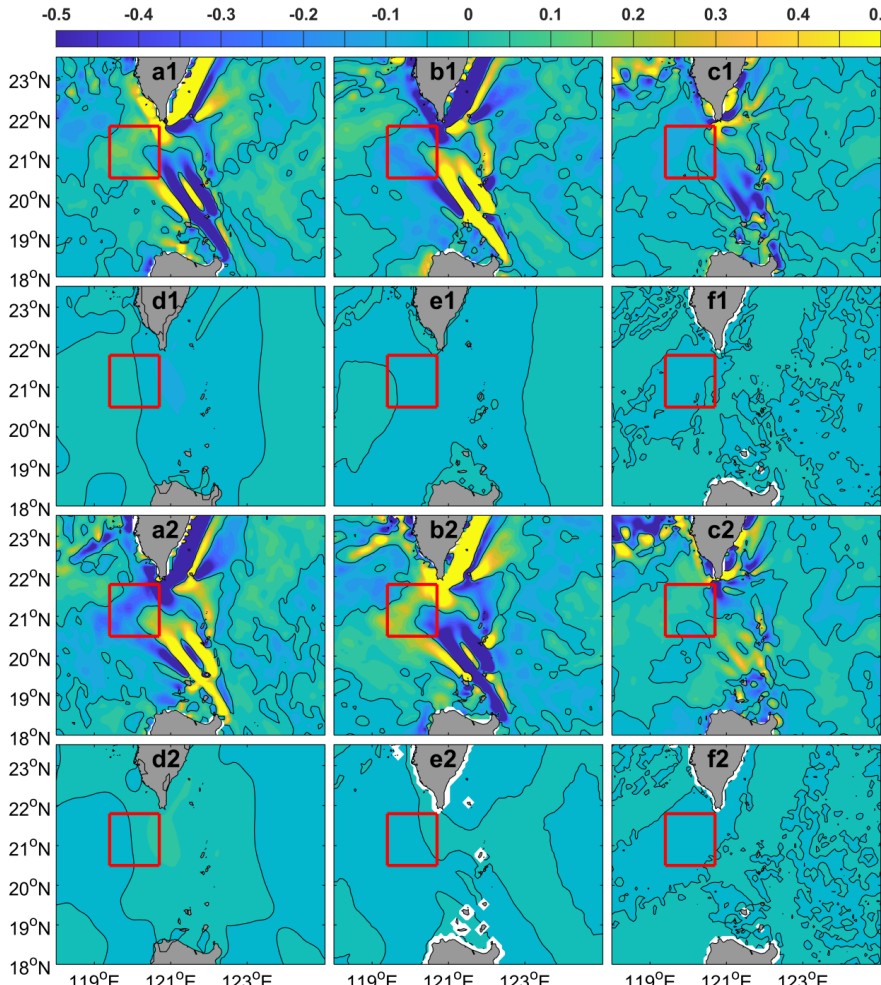


Figure 14. Vorticity budget equation for (a1 - f1) the AE mode of counter-rotating eddy pair and
(a2 - f2) the CE mode of the counter-rotating eddy pair. a1 and a2 represent the zonal advection
term; b1 and b2 represent the meridional advection term; c1 and c2 represent the stretching term;
d1 and d2 represent the beta term; e1 and e2 represent the baroclinic term; and f1 and f2 represent



the diffusion term. The unit is $10^{10}$ s$^{-2}$. The red boxes on the west side of the LS border
20.5-21.8°N, 119.4-120.7°E, and they represent the location of the CE or AE on the west side of
the LS. The black solid line represents the zero contour. This figure is based on the HYCOM data.

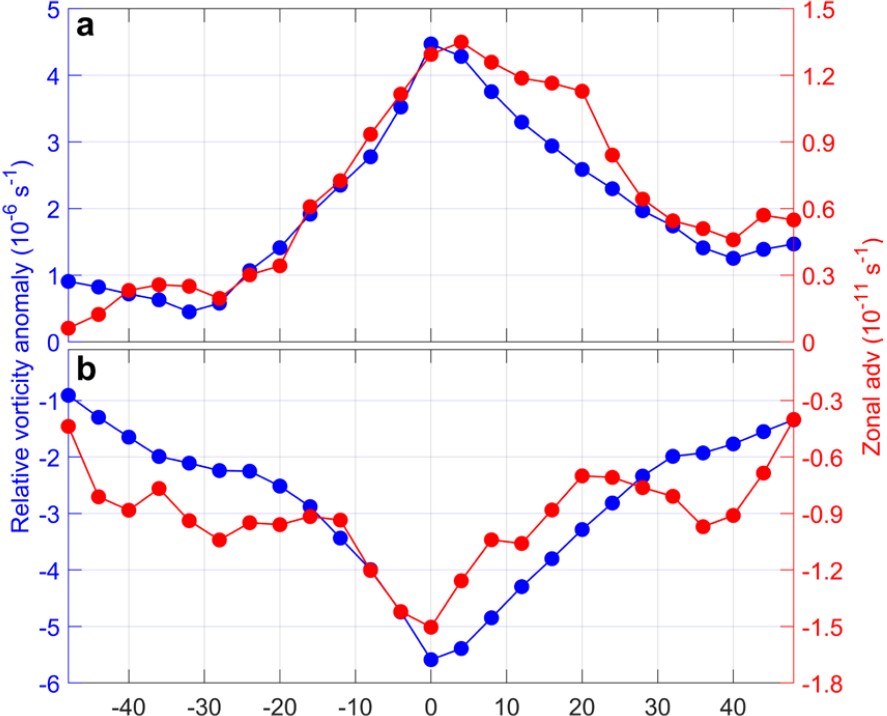


Figure 15. The distribution of the relative vorticity anomaly and the zonal advection of vorticity
surrounded by the red boxes in Figure14 for (a) is for the AE mode of the counter-rotating eddy
pair in the LS; and (b) is for the CE mode of the counter-rotating eddy pair in the LS.

Above, we have proposed that the horizontal velocity shear caused by the mesoscale eddy on

the east side of the LS is transported westward through zonal advection, resulting in the formation
of a counter-rotating mesoscale eddy on the west side of the LS. Horizontal velocity shear will
inevitably lead to barotropic instability. Now, we will verify our conclusion from the perspective
of energy. Figures 16a, 16b, and 16c show that compared to the BC and WW values, the BT




values in the LS are large and most of the values are positive, especially in the area surrounded by
the red box in Figure 16a, which is the junction of the AE and CE. This means that the BT plays
the most important role in the formation of the AE on the west side of the LS.

Figures 16d, 16e and 16f show the BT, BC and WW corresponding to the AE mode of the

counter-rotating eddy pair in the LS, respectively. Its description and dynamic mechanism are
similar to the CE mode of the counter-rotating eddy pair in the LS, so we will discuss the details in
this paper.

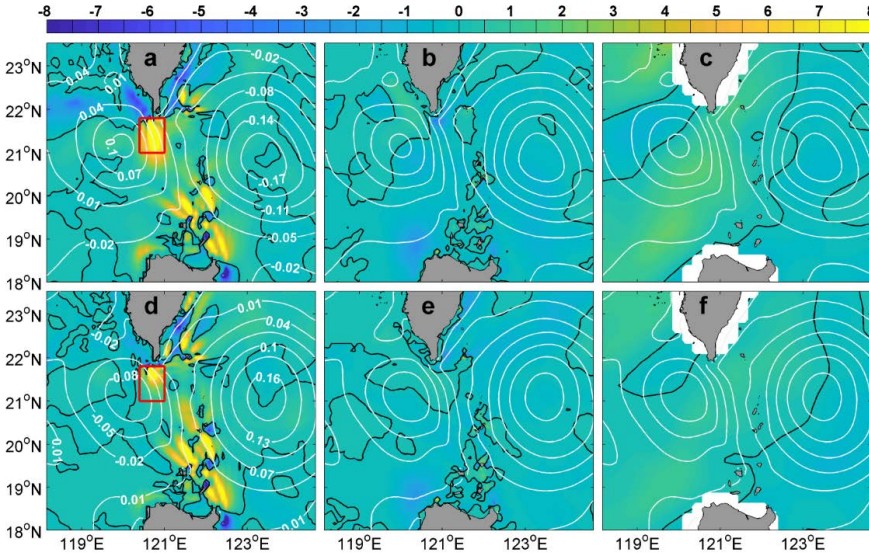


Figure16. (a) BT based on HYCOM data ($10^{-5}$ m$^3$s$^{-3}$) represented by the colors; (b) BC based on
HYCOM data (m$^3$s$^{-3}$) represented by the colors; (c) WW based on CMEMS surface velocity data
and NCDC wind data (m$^3$s$^{-3}$) represented by the colors; The red box borders 21°N-21.8°N,
120.4°E-121°E. The white contours represent the contours of the SSHA. a, b, and c are for the CE
mode of the counter-rotating eddy pair in the LS. d, e and f are for the AE mode of the
counter-rotating eddy pair in the LS.
**4 Discussion and conclusions**



In this study, based on satellite observation data and HYCOM reanalysis data, the
counter-rotating eddy pair in the LS was investigated. The phenomenon of counter-rotating eddy
pair was defined as the stage when an AE (a CE) in the NWP gradually approached the northern
LS, and a CE (an AE) formed on the west side of the LS. This phenomenon exhibited obvious
seasonal variation, that is, the AE mode mainly occurred in the summer half of the year, while the
CE mode mainly occurred in the winter half of the year. The mean durations of the AE mode and
CE mode were both about 70 days. The Leap and Loop patterns of the Kuroshio contributed to the
occurrence of the AE mode and CE mode of the counter-rotating eddy pair, respectively. Based on
energy analysis and the vorticity budget equation, the dynamic mechanism of the occurrence of a
counter-rotating eddy pair is as follows. The AE (CE) in the NWP causes a positive (negative)
vorticity anomaly through horizontal velocity shear on the west side of the LS, and the positive
(negative) vorticity anomaly is transported westward by the zonal advection of the vorticity,
finally leading to the formation of a CE (AE) on the west side of the LS. This conclusion is also
verified by barotropic instability based on the energy analysis.
However, the research presented in this paper is preliminary and some problems require
further study. The occurrence probability of a counter-rotating eddy pair in the LS needs to be
determined. The counter-rotating eddy pair phenomenon involves temporal-spatial variations in
two mesoscale eddies on both sides of the LS, and it is difficult to provide a quantifiable definition
of this phenomenon for a single event. For example, how far apart do the mesoscale eddies on the
east and west sides of the LS need to be in order to define them as a counter-rotating eddy pair. We
preliminarily calculated that the incidence of this phenomenon was about 5%.
Another problem involves threshold of the NWP mesoscale eddies entering the SCS, and





what role does the Kuroshio plays in the counter-rotating eddy pair phenomenon in the LS. When
we illustrated the counter-rotating eddy pair phenomenon in this study, we eliminated the mean
current field, which means that the influence of the Kuroshio was eliminated. However, the role of
the Kuroshio in the energy transfer is still worthy of further study. Numerical simulations can be
useful to address this issue. Our study provides a new perspective on the energy exchange between
the SCS and the NWP.

**Acknowledges**
The authors would like to acknowledge several data sets used in this paper. Satellite remote
sensing geostrophic current data and sea level anomaly were obtained from the CMEMS
(https://resources.marine.copernicus.eu/?option=com_csw&view=details&product_id=SEALEVE
L_GLO_PHY_L4_REP_OBSERVATIONS_008_047), the HYCOM reanalysis data were
downloaded from HYCOM organization (https://www.hycom.org/dataserver/gofs-3pt0/
reanalysis), the data set of wind was provided by National Climate Data Center (https://www.
ncdc.noaa.gov/data-access/marineocean-data/blended-global/blended-sea-winds). Sea surface
temperature data comes from Remote Sensing System (http://www.remss.com/measurements/
sea-surface-temperature/). This study was supported by National Natural Science Foundation of
China (grant number 41806019), National Key R&D Program of China (2019YFD0901305), Key
R&D projection of Zhejiang Province (2020C03012), National Natural
Science Foundation of China (No.41776012), State Key Laboratory of Tropical Oceanography
(South China Sea Institute of Oceanology Chinese Academy of Sciences) open topics (grant
number LTO2011), Key R&D project of Guangdong



Province(2020B1111030002), Major science and technology project of Sanya YZBSTC (SKJC-KJ
-2019KY03). We thank LetPub (www.letpub.com) for its linguistic assistance during the
preparation of this manuscript.

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
