# Peer review of "Counter-rotating eddy pair in the Luzon Strait"

_Ocean Science, 2021_

## Referee Comment (RC1)

**REVIEW**

**Manuscript ID:** os-2021-116
**Title:** Counter-rotating eddy pair in the Luzon Strait
**Author(s):** Sun Ruili, Li Peiliang, Gu Yanzhen, Zhai Fangguo, Yan Yunwei, and Li Bo

**Reviewer's comments:**

Based on satellite remote sensing observation data and Hybrid Coordinate Ocean Model re-analysis, the work studies a counter-rotating eddy pair in the Luzon Strait. The study proposes a mechanism explaining the formation of the counter-rotating eddy pair and observes its seasonal variation over a period of time long enough to classify the pair as a persistent phenomenon rather than a transient regime. The findings are backed by the results and can be a valuable contribution to the field. However, there are some reservations which prevent me from recommending this study for publication in its present form. My overall impression is that the authors did invest a lot of work in this research but cut short on explaining many details without which it makes it difficult to read and understand. I recommend a major revision in hopes that the authors can significantly improve the manuscript.

- The Introduction is lengthy, and presented in a way like it is irrelevant to the study. Discussing the results of some works on the Luzon Strait without explaining how this study complements and/or extends these works, and how the previous results are related to the study looks like it is unimportant for the authors to highlight the position of their work in the state of the art. I would recommend to take the opposite approach.

- The motivation behind the work is unclear. The reader might think of the counter-rotating eddy pair as being a local phenomenon of minor importance. The authors should discuss why they study the counter-rotating eddy pair, why this study is important, and how it contributes to the state of the art.

- The English language has to be improved. In many places, the authors should use Present Indefinite instead of Past Simple (see, e.g. L14-20).

- Remove all web-links from the text and put them in the References.

- Explain how the eddies in Figure 2 as well as the counter-rotating pair have been extracted from data, otherwise it feels like you take a neighbourhood around some local extrema.

- Some figures show fields without explaining whether it is a snapshot, time-mean, or something else; see, e.g. Figures 1 and 3.

- Provide a colorbar and units for Figure 2.

- L109: ... to present, ... What is present? Be specific.

- L121: The wind data was provided by the NCDC. What is the rationale for referring to (Zhang et al., 2006) in line 125?

- L150: "The overbar denotes time averaged" $\rightarrow$ "The overbar denotes a time average (or a time mean) over 70 days". Adjust the following text accordingly.

- Explain in detail how you calculated the period of the counter-rotating eddy pair. Did you extract it from the Fourier analysis of the SSHA time series?

- One reference in line 135 is enough, remove Zhang et al., 2015; Zhang et al., 2017.

- Explicitly define deviations (the primes) in (1)-(3).

- L144: "Where" → "where"

- Give a reference for (4).

- L159: "Where" → "where"

- Do not define the variables in (4) that have already been defined above.

- All the constants in (1)-(4) have to be defined, give the values used in the study.

- Provide a formula for the calculation of time series of the SSHA.

- L183: Remove "in order to obtain a time series".

- Are Figs.4(b)-(c) an average over the positive and negative intensity index, respectively?

- What do you mean by "We counted the temporal distribution of the positive and negative intensity index values." ?

- Explain how you compute RV and RVA in Figs. 8 and 15, respectively.

- Remove produce in line 206.

- Remove "However" in line 382.

---

## Author Response (AR1)

Dear reviewers

Thank you very much for your effort and time to review our manuscript. Your comments and suggestions are very valuable and will be very helpful to improve our manuscript. We have revised the manuscript according to your comments and suggestions. Here are point-to-point replies. The manuscript has been polished again by a professional polishing company (LetPub, www.letpub.com). These replies are divided into two parts. The first part is the author's response for Referee1 and the second part is the author's response for Referee2.

**Part1: Author's response for Referee1**

**Question1**, "The Introduction is lengthy, and presented in a way like it is irrelevant to the study. Discussing the results of some works on the Luzon Strait without explaining how this study complements and/or extends these works, and how the previous results are related to the study looks like it is unimportant for the authors to highlight the position of their work in the state of the art. I would recommend to take the opposite approach."

**Answer:** Thanks. We have made a lot of changes to the introduction in order to shorten the introduction and to emphasize the connection between this study and previous studies, and its importance in this field, which has been highlighted in the introduction of the revised manuscript. We would like to pick several important parts to explain these changes: (1) We give the connection between this study and previous studies from line 52 to line 57. They show that previous studies have studied eddies-eddies interaction phenomenon in the vicinity of the Luzon Strait, however, it is unclear whether this phenomenon of mesoscale eddies-eddies interaction can occur on the east and west sides of the Luzon Strait and plays an important role in the material and energy exchange between the SCS and the NWP. Therefore, our study extends previous studies; (2) We give the importance of our study from line 63 to line 66. They show that our study first propose the counter-rotating eddy pair phenomenon in the Luzon Strait and creats a new form of material and energy exchange between the SCS and the NWP, which would supplement and perfect the theory of material and energy exchange between the SCS and the NWP.

**Question2**, "The motivation behind the work is unclear. The reader might think of the counter-rotating eddy pair as being a local phenomenon of minor importance. The authors should

discuss why they study the counter-rotating eddy pair, why this study is important, and how it contributes to the state of the art."

**Answer:** Thanks. We have added a description of research motivation in the introduction from line 55 to line 57. Our research motivation is to study that whether this phenomenon of mesoscale eddies-eddies interaction can occur on the east and west sides of the Luzon Strait and play an important role in the material and energy exchange between the SCS and the NWP. The importance and contribution of our studies have been given from line 63 to line 66 and from line 395 to line 397 of the revised manuscript.

**Question3**, "The English language has to be improved. In many places, the authors should use Present Indefinite instead of Past Simple (see, e.g. L14-20)."

**Answer:** We have changed Present Indefinite into Past Simple in many places which have been highlighted in the revised manuscript. The manuscript has also been polished again by LetPub company (www.letpub.com).

**Question4**, "Remove all web-links from the text and put them in the References"

**Answer:** We have removed all web-links from the text and put them in the References, which have been highlighted from line 431 to line 432, from line 444 to line 446, from line 458 to line 459 and in line 433 of the revised manuscript. The format of references will be revised according to the requirements of this journal.

**Question5,** "Explain how the eddies in Figure 2 as well as the counter-rotating pair have been extracted from data, otherwise it feels like you take a neighbourhood around some local extrema."

**Answer:** Sorry. Since we have made a lot of changes to the introduction, the original Figure 2 is no longer needed. We have deleted the original Figure 2.

**Question6,** "Some figures show fields without explaining whether it is a snapshot, time-mean, or some-thing else; see, e.g. Figures 1 and 3."

**Answer:** Thanks. We have added a description of time state in the captions of the Figures 1 and 2 (the original Figure 3), which has been highlighted from line 71 to line 72 and from Line 78

to line 79 in the revised manuscript.

 "Provide a colorbar and units for Figure 2."

**Answer:** Sorry. Since we have made a lot of changes to the introduction, the original Figure 2 is no longer needed. We have deleted the original Figure 2.

**Question8,** "L109: ... to present, ... What is present? Be specific."

**Answer:** We have defined the time span of the data and highlighted it in line 90 of the revised manuscript.

**Question9,** "L121: The wind data was provided by the NCDC. What is the rationale for referring to (Zhang et al., 2006) in line 125?"

**Answer:** Zhang et al. (2006) specifically introduced the wind data provided by the NCDC in this reference, so we set it as a reference. If it's not necessary, we can remove it.

**Question10,** "L150: "The overbar denotes time averaged" → "The overbar denotes a time average (or a time mean) over 70 days". Adjust the following text accordingly."

**Answer:** Thanks. We have revised and highlighted it in line 126 and adjust the following text in the revised manuscript.

**Question11,** "Explain in detail how you calculated the period of the counter-rotating eddy pair. Did you extract it from the Fourier analysis of the SSHA time series?"

**Answer:** From Figure 6 and Figure 8 in the revised manuscript, we can see that the counter-rotating eddy pair phenomenon occurs, develops and disappears from $t = -36$ to $t = 36$, which is about 70 days. We have made some attempts to set the period between 65-80 days, and they will not affect our basic conclusion Therefore, we define this period as about 70 days, which has been highlighted from line 128 to line 131 of the revised manuscript. We also analyzed the power spectrum of this SSHA time series and found that the significant period was about 74 days.

**Question12,** "One reference in line 135 is enough, remove Zhang et al., 2015; Zhang et al.,

2017.**"**

**Answer:** We have removed "Zhang et al., 2015; Zhang et al., 2017." In line 114 of the revised manuscript.

**Question13,** "Explicitly define deviations (the primes) in (1)-(3)"

**Answer:** The primes denote deviations from the average value of 35 days before and after this day, which has been highlighted from line 126 to line 127 of the revised manuscript.

**Question14,** "L144: "Where" → "where""

**Answer:** Thanks, we have revised and highlighted it in line 120 of the revised manuscript.

**Question15,** "Give a reference for (4)"

**Answer:** We have added two references, which are highlighted in line 136 of the revised manuscript.

**Question16,** "L159: "Where" → "where""

**Answer:** We have revised and highlighted it in line 137 of the revised manuscript.

**Question17,** "Do not define the variables in (4) that have already been defined above."

**Answer:** We have removed the defined variables in (4) that have already been defined above, And added a description in line 139 of the revised manuscript.

**Question18,** "All the constants in (1)-(4) have to be defined, give the values used in the study."

**Answer:** $\rho_0, v$ are the constants in the formula (1-4). We have given their values used in the study, which has been highlighted in line 122 and line 138, respectively, of the revised manuscript.

**Question19,** "Provide a formula for the calculation of time series of the SSHA."

**Answer:** We have provided a formula for the calculation of time series of the SSHA from line 146 to line 150 and highlighted them in the revised manuscript.

**Question20,** "L183:Remove "in order to obtain a time series". "

**Answer:** We have removed "in order to obtain a time series" in the revised manuscript.

**Question21,** "Are Figs.4(b)-(c) an average over the positive and negative intensity index, respectively?"

**Answer:** Yes, you are right. The original Figure 4(b)-(c) is the Figure 3(b)-(c) of the revised manuscript. We have highlighted it from line 177 to line 178 in the revised manuscript.

**Question22,** "What do you mean by "We counted the temporal distribution of the positive and negative intensity index values."?"

**Answer:** We intended to make statistics on the occurrence time of positive intensity index and negative intensity index. We have further clarified our intention from line 180 to line 181 in the revised manuscript.

**Question23,** "Explain how you compute RV and RVA in Figs. 8 and 15, respectively."

**Answer:** The original Fig. 8 is the Fig.7 of the revised manuscript. We take Figure 7a in the revised manuscript as an example to explain how we compute RV: we first compose SSHA of the time when the AE mode of the counter-rotating eddy pair reached the pinnacle, which is shown in the Figure 3a. Then we give the SSHA from t = -40 to t = 36 at an interval of 4 days, which is shown in the Figure 6. We calculate the relative vorticity (RV) corresponding to different time points from t = -40 to t = 36 in the Figure 6, in which the spatial calculation range of RV is the red boxes in the east and west of the Figure 2. Thus, we get the RV time series in the Figure 7a.

The original Figure 15 is the Figure 14 of the revised manuscript. We take Figure 14a in the revised manuscript as an example to explain how we compute RVA: the acquisition method of the RVA time series in Figure 14a is the same as the one of the RV time series in Figure 7a, where RVA in Figure 14a refers to the RV minus its climate state average.

**Question24,** "Remove produce in line 206."

**Answer:** Thanks. We have removed it in line 190 of the revised manuscript.

**Question25,** "Remove "However" in line 382."

**Answer:** Thanks. We have removed it in line 376 of the revised manuscript.

**Part2: Author's response for Referee2**

**Question1**, "The text needs to be rewritten with an improved English. For example, instead of the word "material" it is better to use "particle" (in line 30 and …)."

**Answer :** Thanks. We have asked a professional English polishing company (LetPub, www.letpub.com) to polish this manuscript. We have used "particle" instead of "material" in the revised manuscript.

**Question2**, "In line 31 the sentence "The LS comprises three straits …" should change to "The LS is comprised of three straits …""

**Answer:** Thanks. We have revised the sentence and have highlighted it from line 32 to line 33 in the revised manuscript.

**Question3**, "In line 33 the sentence "These complex topographic features can …" should be changed to "This complex topography significantly influences/affects the ocean/dynamic processes …"."

**Answer:** Thanks. We have revised the sentence and highlighted it from line 34 to line 35 in the revised manuscript.

**Question4**, "In line 37 the paragraph "The bifurcation of the Kuroshio …" needs to be rewritten."

**Answer:** Sorry. According to the comment of another reviewer, we have revised the introduction, and this sentence has been removed in the revised manuscript.

**Question5**, "In line 42 it is written that "These mesoscale eddies from the NWP can carry an enormous amount of kinetic energy and can alter …" which should be changed to a sentence like

"These mesoscale eddies from the NWP transfer high kinetic energy and impact the local circulation" which also needs a reference."

**Answer:** Sorry. According to the comment of another reviewer, we have revised the introduction, and this sentence has been removed in the revised manuscript.

**Question6**, "In line 45 the sentence "it is important to …" needs to be rewritten".

**Answer:** Sorry. According to the comment of another reviewer, we have revised the introduction, and this sentence has been removed in the revised manuscript.

**Question7**, "In line 51 authors say that Jing and Li (2003) "speculated", to my knowledge in a scientific study nothing is speculated but is "found". Also the sentence is not understandable which needs to be rewritten."

**Answer:** Yes, you are right. We have rewritten this sentence and highlighted it from line 39 to line 42 of the revised manuscript.

**Question8**, "There are typos in the text (e.g. line 33 instead of the word "straits" it is written "straights"; line 77: instead of past tense of the verb "led to" present tense should be used "leads to")."

**Answer:** Thanks very much. We have changed "straights" to "straits" and highlighted it in line 32 of the revised manuscript. The sentence involving "led to" has been adjusted when we revised the introduction of the revised manuscript.

**Question9**, "The introduction of a scientific paper gives sufficient background information to understand the writers' study. Authors give a brief introduction of the region, ocean processes and eddy activity and a detailed summary of the previous studies but a very brief description of the present study I given and its scientific goal and necessity is missing."

**Answer:** Yes, you are right. We have made a lot of changes to the introduction to emphasize our scientific goal and necessity. Our scientific goal is to study that whether this phenomenon of mesoscale eddies-eddies interaction can occur on the east and west sides of the LS and plays an important role in the particle and energy exchange between the SCS and the NWP, which has been

highlighted from line 52 to line 57 of the revised manuscript. The necessity of this study is explained and highlighted from line 63 to line 66 of the revised manuscript.

**Question10**, "References should be checked accurately. There are papers cited in the text but do not exist in the references section (line 50: Jing and Li, 2003; line 54: Yin et al., 2014; line 62: Zhang et al., 2007; line 76: Huang et al., 2019). References need to be checked carefully and the citations must be included in the references section/list."

**Answer:** We apologize for the mistake caused by our negligence. We have checked references accurately and ensure the citations are included in the reference section of the revised manuscript.

**Question11**, "The authors need to follow Copernicus Marine Service instructions to cite the product correctly (https://help.marine.copernicus.eu/en/articles/4444611-how-to-cite-or-reference-copernicus-marine-products-and-services)."

**Answer:** Thanks. We have followed Copernicus Marine Service instructions to cite the product and highlighted it from line 431 to line 432 in the reference section of the revised manuscript. The format of references will be also revised according to the requirements of this journal.

**Question12**, "Also change the citation for the HYCOM model outputs in the text (e.g. model data is obtained from the HYCOM model output by the Naval Research Laboratory)."

**Answer:** Thanks. We have revised and highlighted it from line 95 to line 96 of the revised manuscript.

**Question13**, "Remove the links from the text and insert them in references section"

**Answer:** we have removed the links from the text and insert them in references section and highlighted them from line 431 to line 432, from line 444 to line 446, from line 458 to line 459 and in line 433 of the revised manuscript. The format of references will be also revised according to the requirements of this journal.

**Question14**, "As written in section "Results" authors explain the method applied for the identification and seasonal variation of the eddy pair. This is not a part of results of the study. The method should be move into section two and described in methods section. The results from eddy as "2.3". The identified eddy pair should be shown in results section as a subsection (i.e. 3.1)."

**Answer:** In method section, we have added a subsection 2.2.3 to give the definition of modes and intensity index of the counter-rotating eddy pair, which have been highlighted from line 142 to line 150 of the revised manuscript. We have split the original subsection 3.1 into two parts: "3.1 Identification of the counter-rotating eddy pair in the LS", and "3.2 Seasonal variation of the counter-rotating eddy pair in the LS", and have also made some minor changes to the writing logic accordingly.

**Question15**, "In the text and in figure captions "Figure x is the same as figure x but …" has been used which is not the way to refer to a figure (figure caption). The authors need to be precise about the captions and while referring to a figure in the text."

**Answer:** We have no longer used this sentence "Figure x is the same as figure x but …", and added a complete caption under the Figures involved in the revised manuscript.

**Question16**, "Findings of the study are not discussed in section 4. Authors do not discuss their findings for meridional and zonal advection role."

**Answer:** Yes, you are right. We have added some discussion of meridional and zonal advection role and highlighted them from line 376 to line 382 in the section 4 of the revised manuscript.

**Question17**, "In line 358 authors say that the details of figure 16d, 16e and 16f which illustrate BT, BC and WW will be discussed but in fact no details are found later!"

**Answer:** Yes, you are right. Sorry, it is a typo. Because the details of the original figure 16d, 16e and 16f are similar to the ones of the original Figure 16a, 16b and 16c, and we have discussed details of the original Figure 16a, 16b and 16c, our intention is not to discuss the details of the original figure 16d, 16e and 16f in this manuscript. We have revised the sentence in line 354 in the revised manuscript.

---

## Author Response (AR3)

Dear reviewers and the handling topic editor,

Thanks very much for your time and effort to review our manuscript. Your comments are very valuable and will be very helpful to improve our manuscript. We have revised the manuscript according to your comments. Here are point-to-point replies.

Question 1, "Line 21: day (d)".

Answer: We have replaced "d" with "day (d)" in the Line 21 of revised manuscript.

Question 2, "Line 46: leading (instead of "led")".

Answer: We have replaced "led" with "leading" in the Line 46 of revised manuscript.

Question 3, "Line 65: fulfill (instead of "perfect")".

Answer: We have replaced "perfect" with "fulfill" in the Line 65 of revised manuscript.

Question 4, "Line 71: Remote Sensing Systems (RSS)".

Answer: We have replaced "RSS" with "Remote Sensing Systems (RSS)" in the Line 71 of revised manuscript.

Question 5, "Line 143: are defined as in ... (instead of "are as defined in ...")".

Answer: We have replaced "are as defined in" with "are defined as in" in the Line 144 of revised manuscript.

Question 6, "Line 347: replace "We proposed above" with a better phrase.".

Answer: We have replaced "We proposed above" with "It was mentioned above" in the Line 348 of revised manuscript.

Question 7, "Line 386: requires (instead of "deserves")".

Answer: We have replaced "deserves" with "requires" in the Line 386 of revised manuscript.